# Molecular Basis beyond Interrelated Bone Resorption/Regeneration in Periodontal Diseases: A Concise Review

**DOI:** 10.3390/ijms24054599

**Published:** 2023-02-27

**Authors:** Khadiga M. Sadek, Sara El Moshy, Israa Ahmed Radwan, Dina Rady, Marwa M. S. Abbass, Aiah A. El-Rashidy, Christof E. Dörfer, Karim M. Fawzy El-Sayed

**Affiliations:** 1Biomaterials Department, Faculty of Dentistry, Cairo University, Cairo 12613, Egypt; 2Stem Cells and Tissue Engineering Research Group, Faculty of Dentistry, Cairo University, Cairo 12613, Egypt; 3Oral Biology Department, Faculty of Dentistry, Cairo University, Cairo 12613, Egypt; 4Clinic for Conservative Dentistry and Periodontology, School of Dental Medicine, Christian Albrechts University, 24118 Kiel, Germany; 5Oral Medicine and Periodontology Department, Faculty of Dentistry, Cairo University, Cairo 12613, Egypt

**Keywords:** inflammation, osteocytes, mesenchymal stem cell, bone regeneration, bone resorption

## Abstract

Periodontitis is the sixth most common chronic inflammatory disease, destroying the tissues supporting the teeth. There are three distinct stages in periodontitis: infection, inflammation, and tissue destruction, where each stage has its own characteristics and hence its line of treatment. Illuminating the underlying mechanisms of alveolar bone loss is vital in the treatment of periodontitis to allow for subsequent reconstruction of the periodontium. Bone cells, including osteoclasts, osteoblasts, and bone marrow stromal cells, classically were thought to control bone destruction in periodontitis. Lately, osteocytes were found to assist in inflammation-related bone remodeling besides being able to initiate physiological bone remodeling. Furthermore, mesenchymal stem cells (MSCs) either transplanted or homed exhibit highly immunosuppressive properties, such as preventing monocytes/hematopoietic precursor differentiation and downregulating excessive release of inflammatory cytokines. In the early stages of bone regeneration, an acute inflammatory response is critical for the recruitment of MSCs, controlling their migration, and their differentiation. Later during bone remodeling, the interaction and balance between proinflammatory and anti-inflammatory cytokines could regulate MSC properties, resulting in either bone formation or bone resorption. This narrative review elaborates on the important interactions between inflammatory stimuli during periodontal diseases, bone cells, MSCs, and subsequent bone regeneration or bone resorption. Understanding these concepts will open up new possibilities for promoting bone regeneration and hindering bone loss caused by periodontal diseases.

## 1. Introduction

Periodontium is considered the mirror that reflects many of the human body’s internal status and secrets [1]. Periodontal diseases are chronic infectious conditions causing irreversible damage to the periodontium and its surrounding bone. Worldwide, severe periodontitis is considered among the most prevalent chronic inflammatory conditions affecting the periodontal apparatus [2]. Many factors are involved in the pathogenesis of periodontal diseases where the primary etiological factor is the microbial biofilm. Other etiological factors include some environmental factors, such as smoking and certain systemic diseases such as diabetes mellitus in addition to genetic risk factors affecting the inflammatory and immunologic response of the host [3]. The relationship or balance between the local microbiota and the host’s immune reaction plays a major role in the development and progression of periodontitis [4].

Periodontitis is principally initiated and sustained via dental plaque and the oral microbial biofilm in addition to dysbiosis occurring in the periodontium, causing an extensive release of cytokines, chemokines, and some matrix-degrading enzymes originating from the gingival tissues, the infiltrating immune cells, and the fibroblasts. The net effects of such destructive mediators lead to tissue breakdown, loss of periodontal attachment, and subsequently complete tooth loss [5].

The acute innate immune response during periodontal inflammation depends on a variety of immune cells, such as natural killer cells, neutrophils, and macrophages, besides other variable cytokines. Neutrophils represent the first wave of immune cells within 24 h after the microbial attack. Monocyte–macrophages lineage cells represent the second wave of inflammatory cells peaking at 48–96 h [6].

In an attempt to damage the proteins present in the membranes of some bacteria, neutrophils release elastase; however, this results in the breakdown of type I–IV collagen and the elastin of the periodontal ligaments. Elastase is also able to degrade the extracellular matrix (ECM) by accelerating matrix metalloproteinases (MMPs) cascades, all of which result in attachment loss and pocket formation [7]. The breakdown of the tissues is further exacerbated by neutrophils through the release of MMPs and reactive oxygen species (ROS) [8]. In addition, activated neutrophils increase the expression of receptor activator of nuclear factor kappa-B ligand (RANKL), resulting in a stimulation of osteoclastogenesis and subsequently bone resorption [8,9]. Activated neutrophils release chemokines mediating chemotactic recruitment of T helper (TH)-1 and TH17 cells. TH17 augments the recruitment, the activation, and hence the survival of neutrophils through the release of cell-derived cytokines, for instance, interleukin (IL)-17, interferon-γ (IFNγ), CXC-chemokine ligand 8 (CXCL8; also known as IL-8), granulocyte–macrophage colony-stimulating factor (GM-CSF), and tumor necrosis factor (TNF). Activated neutrophils also secret cytokines capable of mediating the survival, differentiation, and maturation of B cells [8].

Activated B and T cells are also a source of RANKL, resulting in the resorption of bone present in the area of diseased gingival tissues. Additionally, when B cells are activated, this results in the proliferation, differentiation, and maturation of plasma cells. Plasma cells can produce some cytokines such as TNF-α, transforming growth factor-β (TGF-β), IL-6, and IL-10. TNF-α induces the expression of MMPs, thus promoting MMPs-mediated periodontal tissue destruction [10].

Once, the monocyte–macrophage system is activated by the periodontal bacteria and their products, production of huge proinflammatory cytokines occurs [11]. Two distinct phenotypes of macrophages exist: the proinflammatory (M1) macrophages and the anti-inflammatory (M2), arising from the polarization [6]. The M1 and M2 transition is a significant mechanism for the transition between the active periodontitis and the inactive type. Macrophage polarization into M1 is triggered by various stimuli, including microbial stimuli such as IFN-γ and lipopolysaccharide (LPS). M1 produces various proinflammatory factors and represents the primary sources of IL-6, IL-1β, TNF-α, prostaglandin E2 (PGE2), MMP-1, and MMP-2, resulting in the degradation and destruction of the periodontal connective tissues followed by bone resorption [11]. Figure 1 summarizes the role of inflammatory cells in periodontal tissue destruction.

Thus, the relationship between immune surveillance, which is the processes permitting the cells of the immune system to look for and to identify foreign pathogens, and the oral microbe-induced host immune response dictates the progress of the periodontal disease. If a mild host immune response and a local stimulation are balanced, there will be immunological surveillance with a proper immune response. Though, if the pathogenicity of the local microbiota increases due to the colonization of keystone pathogens that overactivate the host immune response, the initiation of tissue destruction arises. Yet, the detailed pathogenesis and immune response to periodontitis remain uncertain and still have many controversies [12].

## 2. Bone Cells

Bone tissue depends on the activity of three main types of cells: osteoblasts, osteocytes, and osteoclasts. Osteocytes make up about 95% of the total cell population [13]. The osteoblast is considered the anabolic part of the previously mentioned cell triad. Osteoblasts deposit osteoid tissue, which is a newly synthesized ECM, consisting of collagen type 1, water, and proteoglycans. This osteoid is then mineralized with hydroxyapatite crystals to obtain the normally required stability [14]. Osteoblasts are derived from skeletal stem cells (SSC), which is a subpopulation of mesenchymal stem cells (MSCs), found in the bone marrow. SSCs can give rise to different cell types, namely, chondrocytes, adipocytes, and osteoblasts [15].

Osteoclasts are large (50–100 nm) multinucleated cells originating from hematopoietic stem cells rather than MSCs [16]. They are resorptive cells, having a ruffled border on the bone matrix-facing side to increase the surface area. This ruffled membrane contains a high number of H+-ATPases in order to lower the localized pH to 4.5, which is needed to dissolve the chemical bonds of calcium in the bone matrix. Osteoclasts attach to the osseous tissue by integrins to ensure a tight seal around the area of low pH. To break down proteins, mainly collagen type I, osteoclasts primarily secrete Cathepsin K and MMPs. This is followed by endocytosis of organic and inorganic fragments [17,18].

Osteocytes are the mature phenotype of osteoblasts. Osteocytes get trapped in the synthesized matrix, locating themselves in the lacuno-canalicular system which is filled with bone fluids. Osteocytes possess 50–60 cellular processes radiating from the cell body and extending through canaliculi, all buried in the mineralized bone matrix [19]. Although osteocytes do not undergo mitotic division, have a slow metabolism, and are anchored to their surroundings, they have a great impact on bone metabolism. They form contact points with each other, osteoblasts, and osteoclasts, spanning a network over the entire bone. This way osteocytes are able to conduct bone turnover [16] (Table 1).

## 3. The Effect of the Periodontal Inflammatory Milieu on Stem Cells

Due to their specific properties, including stemness, proliferation, migration, multilineage differentiation, and immunomodulation, stem cells have been considered a viable strategy to treat tissue damage caused by inflammation [20,21,22,23,24]. In periodontitis, stem cells can be transplanted or recruited to the infection site, acting as key regulators for the inflammatory and immune responses, accelerating the regeneration process [25,26].

Infected tissue-derived stem cells have normal stem cell properties, including reduced immunogenicity and immunosuppression [21]. Dental mesenchymal stem cells (DMSCs), nondental stem cells, and induced pluripotent stem cells are all possible stem cell candidates for periodontal regeneration. In this review, we will focus on DMSCs, specifically those associated with the periodontal tissues, periodontal ligament stem cells (PDLSCs), gingival mesenchymal stem cells (GMSCs), stem cells from the apical papilla (SCAP), and dental pulp stem cells (DPSCs) [25]. The interaction of stem cells and immune cells in the inflammatory milieu may be radically different from that in a healthy state. Maintenance of stemness, colony formation, greater proliferation rate, multilineage differentiation capacity, decreased immunogenicity, and immunosuppression are all features of inflamed stem cells [20].

PDLSCs harvested from inflamed periodontal ligaments (iPDLSCs) constitute a convenient source for stem cells for periodontal regeneration. iPDLSCs are characterized by their enhanced proliferation and migration potential; however, they showed reduced osteogenic potential [27,28] and reduced immunosuppressive properties [29]. They were shown to express high levels of IFN-γ, TNF-α, IL-2, and indoleamine 2,3-dioxygenase (IDO) and low expression of IL-10 [30]. iPDLSCs implanted on collagen sponges successfully formed periodontal-like tissues upon implantation in rats, demonstrating their potential in periodontal regeneration [31]. Similarly, stem cells isolated from inflamed dental pulp (iDPSCs) showed a regenerative potential for periodontal tissues. Additionally, the iDPSCs retained surface marker expression, proliferation, and multilineage differentiation potential as compared to those isolated from healthy pulp [32,33]. iDPSCs impregnated on β-tricalcium phosphate stimulated the formation of alveolar bone in root furcation defects [34]. Periodontal-affected DPSCs and GMSCs showed a high proliferation rate, higher osteogenic potential, and higher calcification deposits than the healthy group. Additionally, proinflammatory cytokines caused cytoskeleton remodeling, which is thought to drive enhanced acquisition in the inflammatory environment, proving their potential in periodontal regeneration [35]. The effect of TNF-α on the osteogenic potential of inflamed SCAPs was assessed in mice’s athymic nude [36]. TNF-α demonstrated a potent inhibitory effect of the bone morphogenic protein (BMP)-9-mediated osteogenic potential of inflammable SCAPs, as well as suppressing ALP, OPN, and OC activities. These inhibitory effects were partially counteracted by high levels of BMP-9, highlighting the potential therapeutic effect of BMP-9 in bone regeneration resulting from chronic inflammation conditions [36].

## 4. Cytokines Effect on Osteoblastogenesis in Periodontal Inflammation

TNF-α, IL-1, IL-6, and IFN-γ, the most efficient proinflammatory cytokines during periodontitis, were reported to exert critical effects on the immunomodulatory abilities of stem cells and their subsequent role in bone remodeling [37,38,39].

### 4.1. IL-1

IL-1 has a contradictory effect on osteogenesis and bone formation. MSC development into osteoblasts and subsequent mineralization has been reported to be aided by IL-1, primarily through the noncanonical Wnt-5a/Ror2 pathway [40]. At doses ranging from physiologically healthy to those found in chronic periodontitis, IL-1β plays a dual role in the osteogenesis of PDLSCs. Low doses of IL-1β enhance osteogenesis of PDLSCs by activating the BMP/Smad signaling pathway; however, higher doses of IL-1β decrease osteogenesis by activating the NF-κB and mitogen-activated protein kinases (MAPK) signaling pathways, meanwhile, suppressing BMP/signaling. PDLSCs with poor osteogenesis release more inflammatory cytokines and chemokines, driving macrophage chemotaxis and highlighting the function of PDLSCs in the etiology of periodontitis [41]. By upregulating the Wnt signaling pathway antagonists DKK1 and sclerostin, IL-1 can effectively suppress osteoblastogenesis [42]. In a highly inflammatory environment, IL-1 levels have also been linked to osteoclastic bone loss [43]. To imitate in vivo periodontitis-inflammatory milieu, GMSCs isolated from free gingival tissues of Sprague–Dawley rats were treated with *Porphyromonas gingivalis* lipopolysaccharides (*P. gingivalis*-LPS) (10 μg/mL). To counteract the deleterious effects of LPS, different doses of IL-1 receptor antagonist (IL-1ra) (0.01–1 μg/mL) were utilized. Cell counts, clone formation rate, cell migration rate, proinflammatory cytokine production, osteogenic differentiation-associated protein/mRNA expressions, and mineralized nodules were found to be inhibited in a time-dependent manner in response to *P. gingivalis*-LPS therapy, a condition that was significantly reversed by dose and time-dependent IL-1ra treatment. TLR4 and IkBα (cellular protein that inhibits the NF-κB) mRNA expressions were also significantly reduced when IL-1ra was added to the LPS-induced media. TLR4/NF-κB activation was similarly reversed by IL-1ra, as evidenced by western blot [44].

### 4.2. IL-6

IL-6 plays a role in the modulation of osteoblastogenesis. DPSCs isolated from healthy pulp and further treated with IL-6 showed significantly increased osteogenic differentiation and increased expression of osteoblasts markers OC and runt-related transcription factor 2 (RUNX2) [45]. IL-6 and its soluble receptor (sIL-6R) potentiated ascorbic acid-mediated osteoblastic differentiation of periodontal ligament cells with upregulation of RUNX2 and ALP activity through insulin-like growth factor production in periodontal ligament cells [46]. IL-6 and sIL-6R also significantly increased osteogenic differentiation and ALP activity of MSCs [47]. Levels of IL-6 and IL-6R significantly increased during bone marrow MSC osteogenic differentiation with a positive correlation to osteogenic differentiation of stem cells. IL-6/IL-6R interaction can induce osteoblastic differentiation of bone marrow MSCs via activation of the signal transducer and activator of the transcription (STAT3) pathway [48]. Conditioned medium of osteocytes treated with IL-6 stimulated upregulation of osteoblastic late marker OC in osteoblasts cell culture [49].

Other reports demonstrated a negative effect of IL-6 on osteoblasts differentiation. IL-6 was associated with downregulation in osteoblasts-related genes, including RUNX2, osterix (OSX), and OC in vitro through activation of Src homology 2 (SHP2)/MAPK kinase/extracellular signal-regulated kinase (ERK), Janus kinase (JAK)/STAT3, and SHP2/phosphoinositide 3-kinase (PI3K)/Akt2 signaling pathways [50]. IL-6 also inhibited osteoblast-calcified nodule formation and ALP activity [51]. Overexpression of IL-6 in mice significantly reduced osteoblastogenesis and increased osteoclastogenesis [52]. Evidence suggests that IL-6 might have a dual role in osteoblasts modulation.

### 4.3. IL-10

Infection-stimulated bone resorption can be suppressed in vivo by IL-10. IL-10-deficient mice exhibited osteopenia, reduced bone formation, and mechanical fragility of the long bones [53,54]. In addition, IL-10 has been shown to enhance bone formation and speed up the healing of bone fractures. IL-10 (10 or 20 nM) increased the metabolic switch from glycolysis to oxidative phosphorylation in DPSCs, whereas IL-10 (5 and 50 nM) had no effect on osteogenic differentiation. The oxidative phosphorylation inhibitor impeded the IL-10-induced enhancement of osteogenic differentiation. These findings reveal that IL-10 can boost DPSC osteogenesis by activating oxidative phosphorylation [55].

### 4.4. IL-17

IL-17 also has a dual role in osteoblasts regulation. While some evidence suggests that IL-17 upregulates osteoblastogenesis and can protect the alveolar bone against periodontitis-mediated bone resorption, others suggest that IL-17 is associated with the downregulation of osteoblastogenesis and may therefore contribute to periodontitis-associated alveolar bone resorption. Interaction of IL-17 with its receptor upregulated osteogenic differentiation of MSCs. Coculturing with osteocytes demonstrated a synergistic effect [56]. It also upregulated expression of ALP, OC, RUNX2, and RANKL expression, meanwhile, reducing mineralization in vitro, indicating that it can exert its influence during the early stages of osteoblastofiggenesis [57]. Other reports demonstrated a positive effect of IL-17 on both early and late osteoblastic differentiation [58,59]. IL-17 was also associated with increased osteoblasts differentiation, mineralization, and proliferation in addition to osteoblast-mediated osteoclastogenesis in vitro and lamellar bone formation in rats with clavarial defects [60].

On the other hand, IL-17 inhibited BMP-2-induced osteoblastogenesis and was associated with reduced expression of ALP, OC, RUNX2, and OSX expression in vitro [61,62]. Treatment of PDLSCs with IL-17 significantly downregulated their osteogenic potential through activation of ERK1,2 and c-Jun N-terminal kinase (JNK) MAPK, implying their role in periodontal-associated bone destruction [63] and inhibited bone formation in rats [62].

### 4.5. TNF-α

However, TNF-α is well-known to inhibit bone formation; its dual function has been proven. TNF-α can either suppress or promote osteogenesis depending on its dose, cell type, and exposure time [6]. TNF-α can favor osteogenic differentiation via NF-κB through increasing expression of BMP-2, OSX, RUNX2, OC, and Wnt signaling pathways [64]. It has been authorized that (10 ng/mL) TNF-α treatment for 0, 3, 5, 15, 30, 60, and 120 min activated the NF-kB pathway during the osteogenic differentiation of DPSCs. Furthermore, TNF-α increased mineralization and expression of BMP-2, ALP, RUNX2, and COL I. It was reported that Pyrrolidine dithiocarbamate, an NF-kB inhibitor, blocked the osteogenic differentiation induced by TNF-α [65].

In the early stages of bone repair, a TNF-α-mediated inflammatory stimulus is mandatory for recruiting osteoblast progenitor cells. This has been confirmed in TNF receptor-deficient mice where only granulation tissue appeared in the marrow cavity on day three after model establishment, while in wild-type mice, young osteoblasts appeared in the marrow space in the same time interval. Moreover, downregulation of type I collagen and osteoclast mRNA expressions has been reported in TNF receptor-deficient mice to 50% as compared to wild-type mice. Endochondral bone formation was downregulated in TNF receptor-deficient mice; however, osteogenesis was not inhibited. On the contrary, intramembranous bone formation was completely absent [66]. TNF-α and its receptors are expressed in a biphasic fashion during bone repair. In mouse models, TNF-α concentration peaks 24 h after bone fracture and returns to baseline within 72 h. TNF-α is mostly expressed by macrophages and other inflammatory cells at this time [67,68]. This short TNF-α signaling is thought to trigger the release of secondary signaling molecules and has a chemotactic impact, attracting bone-regeneration cells. Approximately 2 weeks later, during endochondral bone formation, TNF-α levels rise again. TNF-α is produced by osteoblasts and other mesenchymal cells during this time, including hypertrophic chondrocytes undergoing endochondral bone formation [67,69,70].

TNF-α enhanced osteogenic differentiation and matrix mineralization in MSCs in vitro in a dose-dependent manner [66,71]. Additionally, RUNX2 and OSX levels were downregulated in cell cultures treated with TNF-α at high dosages [72]. It has been reported that the proliferation of human PDLSCs was significantly upregulated following treatment with 10 ng/mL TNF-α; however, ALP enzyme activity and alizarin red mineralization nodule size were significantly reduced following TNF-α treatment for 7 or 21 days. Moreover, the gene and protein expression levels of osteogenic differentiation markers, including RUNX2, OC, and COL-1, were significantly downregulated [73]. These results are contrary to Feng et al. who reported no effect of TNF-α treatment up to 2 h on the proliferation of DPSCs or the cell cycle [65]. Long-term treatment with TNF-α induced inhibitory effects on the in vitro mineral nodule formation of MSCs [72]. TNF-α also stimulates MSC proliferation and immunosuppression via the NF-κB pathway [74], while its inhibitory effect on osteoblast development is mediated by increased production of DKK-1 and Wnt signaling pathway antagonists [75].

Additionally, the infected microenvironment has been reported to affect DMSCs. *P. gingivalis*-LPS, for example, greatly increased the cellular proliferation of DMSCs [76]. Furthermore, coculturing PDLSCs with IL-1β/TNF-α may boost their proliferation rate [77]. On the contrary, *P. gingivalis*-LPS and *Escherichia coli*-LPS, in particular, impede PDLSC osteoblastic differentiation [76,78].

In the absence of BMP-2, an osteogenic supplement, TNF-α and LPS had no effect on the expression of osteogenic markers by human bone marrow MSCs. TNF-α and LPS, on the other hand, increased ALP activity and subsequent matrix mineralization in osteogenic differentiation media or in combination with BMP-2. Both mediators greatly boosted matrix mineralization in preosteoblasts regardless of culture conditions. As a result, it was concluded that both inflammatory factors significantly boost MSCs’ osteogenic potential as well as MSCs that have committed to the osteogenic lineage [79]. Furthermore, preconditioning murine MSCs for 3 days with TNF-α and LPS and then coculturing with macrophages increased anti-inflammatory M2 macrophage marker expression (Arginase 1 and CD206) while decreasing inflammatory M1 macrophage marker expression (TNF-/IL-1ra). MSC immunomodulation of macrophages was dramatically boosted when compared to single treatment controls or a combination of IFN and TNF-α. The only MSCs that displayed enhanced osteogenic differentiation, including ALP activity and matrix mineralization, were those that were preconditioned with LPS and TNF [80].

### 4.6. IL-1β and TNF-α

No change in the surface markers has been reported for the PDLSCs and GMSCs within the IL-1β and TNF-α-inflamed microenvironment; however, when the IL-1β and TNF-α stimulation exceeds a specific level, their favorable effect on cell proliferation and recruitment may be weakened or possibly lead to stem cell death [25,81]. Despite this, transient and low levels of proinflammatory cytokines and microbial pathogens may be involved in the differentiation potential of DMSCs [81]. In the local periodontal environment, IL-1β and TNF-α are responsible for suppressing PDLSC osteogenesis by boosting the canonical Wnt/-catenin pathway and blocking the noncanonical Wnt/Ca2+ pathway [28].

### 4.7. IFN-γ

MSCs’ proliferation and immunomodulatory functions have been found to be stimulated by IFN-γ [82]. Low doses of IFN-γ enhance stem cells’ antigen-presenting activities, minimizing their lysis. High doses, reciprocally, would have the opposite impact [83,84]. For bone marrow MSCs to exert their immunosuppressive effect on T lymphocyte proliferation, IFN-γ has been authorized to be implicated. In addition, both LPS and IFN-γ may cause bone marrow MSCs to secrete functional indoleamine 2,3-dioxygenase and IL-10 [85]. Furthermore, IFN-γ is required for MSC commitment to the osteoblastic lineage, which potentiates bone regeneration both in vitro and in vivo [40,86]. Mice with a knockout IFN-γ receptor exhibited a reduction in bone volume with a low-bone-turnover pattern, a decrease in bone formation, a significant reduction in osteoblast and osteoclast numbers, and a reduction in circulating levels of bone formation and bone resorption markers. These data support an important physiologic role for IFN-γ signaling as a potential therapeutic target for bone loss [87]. Other literature has reported that IFN-γ suppresses allogeneic MSC-induced osteogenesis [88]. Through T cell activation, IFN-γ also has a stimulatory effect on osteoclastogenesis and bone loss [89].

### 4.8. TGF-β

TGF-β is a well-known anti-inflammatory cytokine that can help MSCs proliferate and differentiate [69]. Signaling between TGF-β and BMP is involved in the vast majority of cellular functions and is crucial throughout life. TGF-β/BMP signaling is mediated by both canonical Smad-dependent and noncanonical Smad-independent pathways (e.g., p38 and MAPK). Both the Smad and p38 MAPK pathways converge on the RUNX2 gene following TGF-β/BMP activation to govern mesenchymal precursor cell development [90]. TGF-β activated receptors form a complex with Smad4, which translocates from the membrane into the nucleus to interact with RUNX2, resulting in numerous osteogenic genes being activated [91]. High dosages of TGF-β1, on the other hand, hindered bone marrow MSC osteogenesis and slowed bone healing in vivo. The effects of different TGF-β1 levels on osteogenic differentiation and bone repair were inversely proportional. Low TGF-β1 dosages activated Smad3, increased their binding to the promoter of BMP-2, and enhanced BMP-2 production in bone marrow MSCs. At high TGF-β1 levels, BMP-2 production was suppressed by modifying Smad3 binding sites on its promoter. Furthermore, high TGF-β1 doses elevated tomoregulin-1 levels in mice, causing BMP-2 suppression as well as hindering bone formation [92]. The latter study clarifies the conflicting results of the negative relationship between TGF and osteoblastogenesis [58].

### 4.9. Other Cytokines and Osteoblastogenesis

Other cytokines might have an effect on MSCs during in vitro and in vivo bone regeneration and remodeling. IL-22, for example, stimulates MSCs’ osteogenic activity [93]. By stimulating the canonical Wnt-catenin pathway, IL-23 has been shown to induce osteogenic differentiation of MSCs [94]. Furthermore, new evidence suggests that the anti-inflammatory cytokine IL-27 can enhance bone production by inhibiting osteoblast death and suppressing osteoclastogenesis [95].

## 5. Osteoclasts and Osteoclastogenesis in Periodontal Inflammation

### 5.1. Osteoclast Differentiation (Osteoclastogenesis)

Osteoclasts are hematopoietic in origin; they arise through the fusion of multiple monocytes [96,97]. The process of osteoclastogenesis begins with common myeloid progenitor cells, which arise from hematopoietic stem cells within the bone marrow under the influence of factors, including stem cell factors, IL-3, and IL-6. Common myeloid progenitors are stimulated to give rise to granulocyte/macrophage progenitors under the influence of GM-CSF. Granulocyte/macrophage progenitors further differentiate into the monocyte–macrophage lineage, the precursor of osteoclasts, following stimulation by macrophage-CSF (M-CSF) [98,99]. M-CSF is essential for osteoclasts proliferation and survival [100]. The binding of M-CSF to its receptor CSF-1 receptor (c-Fms) activates intracellular PI3K and growth factor receptor bound protein 2 (Grb 2), which in turn activates Akt and ERK signaling in osteoclasts precursor [101]. Within the bone environment, M-CSF arises primarily from osteoblast cells in addition to bone marrow stromal cells [102].

Further differentiation of osteoclasts precursor is mediated by osteoblasts and bone marrow stromal cells through RANK and its ligand (RANKL) through OPG, RANKL, and RANK axis. OPG, RANKL, and RANK are both TNF/receptor superfamily members [103]. RANKL exerts its function through binding to RANK on osteoclast progenitors’ surfaces, resulting in a signaling cascade that eventually promotes differentiation and fusion of osteoclast precursors. It can also promote mature osteoclasts’ survival and activity [100,103].

RANKL is a key player in periodontitis-associated bone resorption. It is expressed by osteoblasts and bone marrow stromal cells [100]; it is also expressed by gingival epithelial cells and fibroblasts [104,105] and periodontal ligament cells [106]. Activated T and B cells can express RANKL in periodontal-diseased gingival tissue which potentiates bone resorption [107,108]. Additionally, cementoblasts have also been shown to express RANKL and can enhance osteoclastogenesis in vitro [109,110].

Upon binding of RANKL to RANK, TNF receptor-associated factors become activated, which results in a cascade of intracellular signaling, eventually activating cascades of adaptors/kinases such as NF-κB and MAPKs, including p38, JNK, and ERK. Eventually, this results in several transcription factors’ activation, including NF-κB, activator protein-1, cyclic adenosine monophosphate response element-binding protein, and nuclear factor of activated T cells 1 (NFATc1), which in turn results in the induction of the expression of osteoclastogenic markers, including tartrate-resistant acid phosphatase (TRAP), a dendritic cell-specific transmembrane protein, osteoclasts-associated receptor (OSCAR), β3 integrin, osteopetrosis-associated transmembrane protein-1, B-lymphocyte induced maturation protein 1, and cathepsin K [101].

The process of osteoclastogenesis is regulated by OPG, which is a decoy ligand to the RANKL receptor. The binding of OPG to RANK can downregulate osteoclast differentiation [103,111,112]. Periodontal ligament cells can regulate osteoclastogenesis through the expression of OPG [106]. Osteoclast differentiation in periodontal diseases is illustrated in Figure 2.

### 5.2. Cytokines and Osteoclastogenesis in Periodontitis

Several cytokines can modulate the process of osteoclastogenesis during periodontal disease with subsequent modulation of the process of alveolar bone resorption. IL-1 super family (IL-1ɑ, IL-1β, IL-33, IL, IL-36), IL-6, IL-8, IL-11, IL-17, IL-22, IL-34, and TNF are proinflammatory cytokines that can upregulate osteoclast differentiation [113,114,115,116,117,118]. Periodontitis-associated bone destruction can be attributed to the upregulation of proinflammatory cytokines that favors bone resorption.

#### 5.2.1. IL-1 Super Family and Osteoclastogenesis

IL-1 was associated with increased osteoclastogenesis in periodontitis. Osteoclast formation and the progression of inflammatory cells toward alveolar bone were significantly reduced upon blocking of IL-1 and TNF in nonhuman primates [119]. Furthermore, periodontal ligament fibroblasts precultured with IL-1 β significantly upregulated osteoclastogenesis in peripheral blood mononuclear cells culture [120]. IL-1 receptor antagonist significantly reduced the number of osteoclasts in an experimental tooth movement rat model [121]. IL-1 can potentiate RANKL-induced osteoclastogenesis [122,123]. Additionally, IL-1 can upregulate RANKL expression by stromal cells to induce osteoclastogenesis through p38 MAPK [115]. IL-1β can potentiate cementoblasts-induced osteoclastogenesis via upregulation of cementoblast RANKL expression [110]; IL-1ɑ was associated with upregulation of RANKL and downregulation of OPG mRNA expression in periodontal ligament cells via the ERK pathway [124].

IL-1 can also induce osteoclastogenesis via an intracellular pathway independent of RANKL/RANK interaction [123]. Binding between IL-1 and its dimeric receptors IL-1R1 on the surface of osteoclast progenitors triggers intracellular signaling cascade and upregulating transcriptional factors, including JNK, P38, and ERK. IL-1 can also upregulate microphthalmia transcription factor, which induces osteoclast-specific genes such as OSCAR and TRAP [123]. However, IL-1 can only upregulate differentiation of osteoclasts precursor in presence of RANKL or TNF-α, which can upregulate IL-1 secretion by stromal cells and can upregulate IL-1RI expression via c-Fos and NFATc1 [115,123]. RANKL is responsible for priming bone marrow macrophage osteoclast genes, including the gene coding for NFATc1, to be responsive to IL-1-mediated osteoclastogenesis [125].

IL-33 is also involved in upregulating osteoclastogenesis. IL-33 can potentiate RANKL-induced osteoclastogenesis [126]; it was involved in the upregulation of RANKL expression by osteoblasts via ERK and p38 MAPK [127] and by periodontal ligament cells in vitro [128]. Further, expression of IL-33 was upregulated in gingival samples from patients with chronic periodontitis and in rats with induced periodontitis and was accompanied by increased RANKL expression in gingival epithelial cells [129]. IL-33 can also upregulate osteoclastogenesis through a pathway independent of RANKL/RANK interaction. IL-33 interacts with its receptor to activate signaling molecules required for osteoclastogenesis, including a spleen-associated tyrosine kinase, phospholipase Cc2, Grb2-associated-binding protein 2, MAPK, TAK-1, NF-kB, IL-33, TNF-α receptor-associated factor 6 (TRAF6), nuclear factor of activated T cells cytoplasmic 1, c-Fos, c-Src, cathepsin K, and calcitonin receptor [126].

On the contrary, other reports demonstrated that IL-33 was associated with a significant reduction in the number of osteoclasts in vivo [130] and can downregulate the process of osteoclastogenesis through impeding osteoclast differentiation factors such as NFATc1 [131], meanwhile, upregulating osteoclasts apoptosis via upregulation of pro-apoptotic molecules, such as BAX, Fas, and FasL [132]. Additionally, IL-33 is highly expressed in chronic apical periodontitis lesions with a negative correlation with RANKL and a positive correlation with OPG expression, suggesting a protective role of IL-33 against bone loss in periodontitis [133]. The conflicting results regarding IL-33’s effect on osteoclastogenesis can be attributed to its local concentration. A high concentration of IL-33 was associated with increased RANKL expression in periodontal ligament cell culture, while lower levels of IL-33 were associated with increased OPG expression, suppressing the process of osteoclastogenesis [128].

IL-36 γ was also suggested to be involved in periodontitis-associated osteoclastogenesis. IL-36 γ expression in the gingiva was upregulated and positively correlated with a RANKL-to-OPG ratio, indicating its role in RANKL-mediated osteoclastogenesis [134].

#### 5.2.2. IL-6 and Osteoclastogenesis

IL-6 interacts with its receptor IL-6R to upregulate the process of osteoclastogenesis [52,135] through upregulating RANKL expression on osteoblasts [113]. Soluble IL-6R in the presence of IL-6 can upregulate osteoclastogenesis in bone marrow cells cocultured with osteoblasts [136]. IL-6R inhibition can block osteoclastogenesis in vivo and in vitro [137] and can significantly reduce alveolar bone resorption in the periodontitis model in rats [138]. IL-6 can also upregulate osteocyte RANKL expression to upregulate osteoclastogenesis [139].

IL-6 can also upregulate osteoclastogenesis via a pathway independent of RANK/RANKL interaction [114]. Both IL-6R and IL-11R are expressed on osteoclasts and can transduce signals via the GP130 signaling pathway [114,140]. IL-6 and IL-11 were both able to induce osteoclastogenesis in CD14^+^ monocytes in the absence of RANKL-expressing cells. Further, osteoclastogenesis was not inhibited by OPG or RANK-Fc (a recombinant RANKL antagonist) but was inhibited by glycoprotein 130 antibodies, denoting that IL-6 can upregulate osteoclastogenesis independent of RANKL [114]. However, the osteoclastogenic effect of IL-6 is dependent on the presence of RANKL in the environment [141].

On the contrary, IL-6 was demonstrated to be nonessential for physiologic bone resorption in vivo but can stimulate osteoblastic bone formation [140]. Moreover, IL-6 was associated with reduced osteoclastogenesis via suppression of NF-κB pathways [142]. A dual role of IL-6 in osteoclastogenesis was revealed. IL-6 and sIL-6R, in the presence of a high level of RANKL, upregulated osteoclast differentiation through the upregulation of NF-κB, ERK, and JNK phosphorylation. While in the presence of low levels of RANKL, IL-6 and sIL-6R downregulated osteoclast differentiation [141], indicating that the level of RANKL is essential for directing the cellular response to IL-6.

#### 5.2.3. IL-17A and Osteoclastogenesis

In the periodontitis rat model, IL-17A was linked to increased alveolar bone loss and a rise in osteoclasts [143,144]. Blocking of IL-17A in rats with induced periodontitis significantly reduced the alveolar bone loss osteoclast number [145]. IL-17A can also upregulate osteoclastogenesis via RANKL upregulation. IL-17A upregulated RANKL expression in osteoblasts [143] and periodontal ligament cells [128] and stimulated osteoclastogenesis via upregulation of the expression of autophagy-related genes and proteins in vitro [143,144]. Periodontitis-associated upregulation of IL-17A and RANKL is mediated through eliciting receptors expressed on myeloid cells-1 (TREM-1). TREM-1 blockage downregulated IL-17A and RANKL expression while OPG expression was upregulated [146].

#### 5.2.4. IL-22 and Osteoclastogenesis

Another IL involved in periodontitis-associated bone catabolism is IL-22. Levels of IL-22 were significantly higher in the gingiva of patients with periodontitis as compared to healthy individuals and were positively correlated with pocket depth [147]. Patients with periodontitis’ tissue homogenates actively promoted osteoclast activity. When IL-22 was neutralized, this effect vanished, showing how it affects osteoclast activity [147]. IL-22 mediates osteoclastogenesis through the upregulation of RANKL expression. Increased levels of Il-22 and RANKL was correlated with increased alveolar bone resorption in experimental periodontal lesions in rats [148]. Additionally, IL-22 upregulated RANKL expression in vitro in human periodontal ligament fibroblasts via the MAPK signaling pathway, indicating their role in osteoclastogenesis [149].

#### 5.2.5. Other Interleukins and Osteoclastogenesis

IL-8, produced mainly by dendritic cells during periodontal inflammation, was implicated in the upregulation of osteoclastogenesis through stimulating inflammatory cells to secrete IFN-γ, IL-17, TNF-α, IL-1β, and RANKL [150]. IL-11 could upregulate RANKL expression by osteoblasts in vitro, thus supporting osteoclastogenesis [151], and could upregulate osteoclastogenesis independent of RANK/RANKL interaction [114]. IL-34 was also linked to increased bone resorption in periodontitis. IL-34 expression in gingival fibroblasts is upregulated upon TNF-α and IL-1β treatment in vitro. IL-34 significantly upregulated osteoclastogenesis in vitro the in presence of RANKL in bone marrow macrophages [152].

#### 5.2.6. TNF and Osteoclastogenesis

TNF acts on osteoblasts [151], stromal cells [153,154], gingival epithelial cells via TNFR1 and protein kinase A signaling [104], and osteocytes [155] to increase RANKL expression. TNF-α can upregulate RANK expression on osteoclasts precursor and sensitize precursor cells to RANKL [154]. IL-1, IL-6, and TNF show a synergistic effect on stimulating osteoclastogenesis [156].

TNF can also downregulate OC, ALP, and RUNX2 expression, impeding osteoblast differentiation [157]. It can also inhibit the WNT signaling pathway to downregulate osteoblast function [158] and increase osteoblast apoptosis [159], resulting in suppression of osteoblast action, which aggravates bone loss in periodontitis. TNF alone cannot initiate osteoclastogenesis; the presence of RANKL is essential for TNF-mediated osteoclastogenesis [115,153,160,161,162] as RANKL prime macrophages undergo osteoclast differentiation in response to TNF [153]. Even though some studies demonstrated that TNF could stimulate osteoclastogenesis independent of the RANK/RANKL pathway [163,164], TNF-α failed to stimulate osteoclastogenesis in RANK-deficient mice [165]. TNF-α and RANKL have a synergistic effect on the process of osteoclastogenesis via the activation of NF-kB and SAPK/JNK [153].

### 5.3. Micro RNA and Osteoclastogenesis in Periodontitis

Micro RNA (miRNA) also plays a role in the regulation of osteoclastogenesis in periodontitis. Several miRNAs, including miRNA-124 [166] and miR-218, can reduce periodontitis-associated bone resorption through the downregulation of MMP 9 [167]. While miRNA-31 was upregulated during the process of osteoclastogenesis under RANKL stimulation, miRNA-31 inhibition suppressed RANKL-induced osteoclastogenesis and was associated with impaired actin ring formation via upregulation of RhoA expression [168]. Treatment of cells with a combination of TNF-α and RANKL effectively altered the expression of 44 microRNAs. miR-378 was upregulated and miR-223 was downregulated, while miR-21, miR-29b, miR-146a, miR-155, and miR-210 were upregulated during osteoclastogenesis upon administration of a combination of TNF-α and RANKL [169]. Cytokines and pathways involved in osteoblastogenesis and osteoclastogenesis in periodontitis are summarized in Table 2.

### 5.4. Bacterial Factors and Osteoclastogenesis in Periodontitis

#### 5.4.1. Periodontal Bacteria

The complex bacterial species responsible for periodontal disease include *P. gingivalis*, *Tannerella forsythensis*, *Treponema denticola*, *Prevotella intermedia*, and *Aggregatibacter actinomycetemcomitans* [170]. Precisely, *P. gingivalis*, *Tannerella forsythensis*, and *Treponema denticola* are identified as the red complexes that have obtained the attention of researchers [171]. The oral administration of *P. gingivalis* [172] and *Aggregatibacter actinomycetemcomitans* [173] resulted in increased alveolar bone resorption which was accompanied by an increase in osteoclasts numbers with a decrease in osteoblasts numbers [174,175]. These results could be explained by the upregulation of RANKL expression in osteoblasts and the downregulation of OPG in vitro induced by *P. gingivalis* [176,177]. Several studies reported that alveolar bone resorption is commonly associated with multiple bacterial infections rather than a single bacterial infection [178,179,180]. Microbe-associated molecular patterns are unique structural components, including LPS and peptidoglycan (PGN), that can elicit an immune response [181]. The recognition of microbe-associated molecular patterns by the host cell occurs through TLR and nucleotide-binding oligomerization domain (NOD)-like receptors [182,183].

#### 5.4.2. Lipopolysaccharides

LPS is a bacterial endotoxin capable of provoking a local immune response [76]. LPS is a major component of gram-negative bacteria’s outer membrane that consists of three parts. The outermost domain is formed of polysaccharide chains (O-antigen), while the innermost domain consists of hydrophobic fatty acid chains (lipid A). The O-antigen is attached to lipid A through the oligosaccharide core [184]. The virulence of LPS is attributed to the release of lipid A, which activates the immune response [185]. Bacterial LPS interacts with TLR-4 expressed on innate immune cells, such as macrophages and dendritic cells [186,187]. Consequently, this interaction promotes the production of different cytokines, such as TNF-α, IL-1, and PGE2 [153,188,189]. These cytokines have an important role in osteoclast progenitor cells maturation and bone resorption as they can stimulate RANKL expression in osteoblasts [151,190].

Moreover, LPS is involved in osteoclast formation. The addition of *Escherichia coli*-derived LPS to murine RAW 264.7 macrophage cells (osteoclast progenitor cell line) induces the formation of osteoclasts with bone-resorbing activity [191]. It was reported that LPS can directly interact with TLR-4 on osteoblast-enhancing RANKL expression [192]. The RANKL/RANK interaction led to the differentiation and activation of osteoclasts. This was confirmed by the inhibition of TLR-4 and TLR-2 expression in mouse-derived osteoblasts, which led to a reduction in RANKL expression upon exposure to LPS [193]. Additionally, LPS is capable of inhibiting RANKL-induced osteoclastogenesis during the early stages of osteoclastic differentiation [194].

LPS-mediated RANKL expression signaling pathways are dependent on the type of bacteria from which LPS originate and their binding with TLR. It was reported that LPS derived from *P. gingivalis* upregulate RANKL expression by activating the JNK pathway and activator protein-1 transcription factor in osteoblasts [177]. In the same way, *Porphyromonas endodontalis* upregulate RNAKL expression through the JNK pathway [193]. On the other hand, *Escherichia coli*-derived LPS upregulate RNAKL expression by activating PI3K signaling molecules or extracellular signal-regulated kinase [193].

Although studies are reporting that LPS alone inhibit osteoclast formation from osteoclast precursors and only promote osteoclastogenesis in RANKL-pre-treated cells [194,195], others reported that LPS induced bone resorption in vitro and in vivo [196,197,198,199]. Therefore, future research addressing the role of LPS in osteoclastogenesis using RANK knockout culture systems and animal models is recommended.

#### 5.4.3. Peptidoglycan

PGN is a polymer composed of sugars in addition to peptides present in gram-negative and gram-positive bacteria. It was demonstrated that the systemic administration of *Staphylococcus aureus* PGN-induced systemic arthritis, evident by bone resorption in vivo [200]. Furthermore, PGN promotes the fusion of osteoclasts during osteoclastogenesis of macrophage-like Raw264.7 cells [201]. Comparable to LPS, PGN also can inhibit osteoclastogenesis in a dose-dependent manner, which indicates that the timing of RANKL stimulation is a critical factor for osteoclastogenesis [194,202,203]. Despite the local administration of *Staphylococcus aureus*, PGN in mice induced alveolar bone resorption while *Escherichia coli* PGN failed to induce bone resorption [204]. PGN induces bone resorption through stimulation of TLR-2 [205]. In addition, PGN fragments stimulate inflammatory responses via NOD1 and NOD2, which are cytoplasmic proteins that sense bacterial byproducts. It was reported that PGN-derived, gram-positive bacteria stimulate NOD2 [206,207,208], while gram-negative PGN is a potent stimulator of NOD1 and a weak stimulator of NOD2 [206,207,209]. Although NOD1 is expressed in most tissues, NOD2 is especially expressed in immune cells, such as macrophages [210,211]. This could explain the increased bone resorption activity associated with gram-positive PGN. Furthermore, the role of PGN during bone resorption in periodontitis was emphasized in *P. gingivalis*-induced periodontitis in NOD2 knockout mice, which resulted in suppression of RANKL expression and impaired alveolar bone resorption [212]. Even though several studies addressed the role of LPS and PGN in osteoclastogenesis in vitro and in vivo, further studies focusing on the exact mechanisms responsible for their effect and how they could be counteracted are needed.

## 6. Osteocytes and Periodontal Inflammation

### 6.1. Crosstalk among Osteocytes, Osteoclasts, and Osteoblasts

The crosstalk among osteocytes, osteoclasts, and osteoblasts is essential for physiological bone turnover and homeostasis maintenance. RANKL is primarily obtained from osteocytes, which acts upon RANK, regulating osteoclast differentiation. Osteocytes express RANKL ten times higher than osteoblasts [213,214]. Moreover, osteocytes are able to control osteoclast formation and bone resorption by upregulating RANKL and downregulating OPG or, in reverse, stimulating the opposite conditions to decrease bone resorption [215].

Osteocytes can exert a stimulatory and inhibitory effect on osteoblasts. Among the most potent signals originating from osteocytes, Sclerostin and DKK1 are strong antagonists of Wnt/β-catenin signaling, which play an important role in promoting osteoblastogenesis and matrix formation [216,217].

### 6.2. Osteocytes in Periodontitis

Periodontitis tissue contains many pathological factors, including biologically active substances in bacterial plaques and inflammatory mediators produced by immune cells as discussed earlier. These factors are able to increase RANKL expression in osteocytes.

LPS from gram-negative bacteria are recognized by TLR2 on the osteocyte surface, which irritates the downstream MAPK/ERK 1/2 signaling pathway and transcription factors, resulting in upregulation of IL-6 expression [218]. IL-6 causes glycoprotein 130-mediated JAK activation, which then phosphorylates STAT [219], that translocates into the nucleus and eventually increases RANKL expression in osteocytes [219,220]. TNF-α binds to the TNF receptor on the osteocyte surface, TNF-α activates the ERK1/2, P38, and JNK/MAPK signaling pathways and/or the NF-κB, which boosts RANKL expression in osteocytes and subsequently promotes alveolar bone resorption [155,221].

Strikingly, RANKL content is significantly increased, accompanied by downregulated OPG levels in periodontitis [222]. As the RANKL/OPG ratio increases, the bone resorption area enlarges by the increased osteoclasts cells number [223]. oRANKL subcellular trafficking regulation in osteocytes were investigated; osteocytes provide RANKL as a membrane-bound form to osteoclast precursors through direct cell-to-cell interaction at the extremities of dendritic processes. OPG acts as a RANKL trafficking regulator by transporting the newly synthesized RANKL to the cell surface when it is stimulated with RANK to regulate osteoclastogenesis [224].

Sclerostin, a secreted glycoprotein generated by osteocytes and encoded by the *SOST* gene, was found to be highly expressed inGCF ofC\chronic periodontitis patients compared to healthy patients [225]. TNF- α enhance Sclerostin expression in osteocytes via an NF-κB-dependent mechanism, where NF-κB binds to the *SOST* promoter region and induces an increase in sclerostin expression [221,226]. Moreover, DKK1, which is an endogenous secretory protein mainly produced by osteocytes, can enhance the TNF-α-induced sclerostin in osteocytes to inhibit osteoblast activity [227]. Sclerostin can be considered an effective therapeutic target for periodontal disease treatment, as alveolar bone volume improvement was noticed using DKK1- and Sclerostin-specific antibodies [228,229].

Both Sclerostin and DKK1can interrupt Wnt/β-catenin signaling and compete with WNT proteins for binding to the extracellular regions of low-density lipoprotein receptor-related protein-5/6 on osteoblasts, thus hindering osteoblastogenesis [230,231]. Moreover, DKK1 can inhibit the Wnt/β-catenin signaling pathway, thereby decreasing OPG expression, which in turn leads to an increase in the local ratio of RANKL/OPG in osteocytes, which increases osteoclastogenesis and promotes bone absorption [232,233].

### 6.3. Osteocyte Senescence in Periodontitis

Aging and osteoporosis are common causes of osteocyte senescence [234]. However, in young individuals, early senescence can occur as a stress reaction to inflammation or extracellular matrix remodeling through the secretion of senescence-associated secretory phenotype proteins, which leads to a state of irreversible growth stagnation [235,236]. It was demonstrated that the advanced senescence of osteocytes is related to periodontal pathogens and their products being in close contact with the alveolar bone for a long period. LPS exposure can cause DNA damage in osteocytes to exacerbate the adaptive immune response and periodontal inflammation [237]. This process is called inflamm-aging and may risk bone remodeling and bone homeostasis [238].

### 6.4. Apoptotic Osteocyte in Periodontitis

In periodontitis, for inflammatory cells and periodontium cells, including osteocytes, apoptosis is amplified. The apoptosis of these cells exerts a significant effect on the progression of chronic inflammation and tissue damage [239,240]. Although it is still debatable, there is an intrinsic and complex cross-talk mechanism between apoptotic osteocytes and osteoclastogenesis. Apoptotic osteocytes produce apoptotic bodies that promote osteoclast progenitor cells to differentiate into osteoclasts. Apoptotic osteocytes can secrete RANKL and directly modulate osteoclast formation and bone remodeling [241,242]. However, this process is not entirely dependent on RANKL, as osteoclastogenesis is not decreased in the presence of OPG concentrations ranging from 50 ng/mL to higher levels (>400 ng/mL). necrosis occurs when apoptotic osteocytes are not removed which aggravates the secretion of multiple cytokines and immune cells aggregation, which potentiates the generation of pro-inflammatory molecules and stimulates the secretion of RANKL in neighboring cells [243].

Mononuclear osteoclast precursor cells produce TNF-α that recognizes apoptotic osteocytes surface markers, leading to enhanced osteoclast formation [244]. Apoptosis of osteocytes may also be aided by bacterial stimulus and inflammatory substances, according to literature [245]. Clarifying the role of osteocyte death in periodontitis will help to improve future clinical prevention, diagnosis, and treatment of periodontitis.

## 7. Clinical Implications

### 7.1. The Potentiality of Using the GCF and the Salivary Biomarkers as Diagnostic Tools for Periodontal Diseases

Some biomarkers could be used as diagnostic tools for the course of many diseases in the human body, including periodontal diseases [246]. Many studies were performed to correlate the presence of one or a group of cytokines in the saliva or the GCF and the periodontal condition of the patient [139,244,247,248,249]. Focusing on the biomarkers present in the GCF, it was proved that IL-1 with both its types, IL-1α and IL-1β, possess great potential in distinguishing periodontitis from periodontal healthy cases [247]. Such biomarkers could reveal improved diagnostic ability when combined with anti-inflammatory cytokine, IFN-γ, and IL-10 [247]. Another group of biomarkers that could be used in the diagnosis of periodontal diseases is IL-1β, IL-8, MMP-13, osteoprotein, and osteoactivin [139]. In addition, the level of IL-17, IL-18, and IL-21 in the GCF could be correlated with the severity of the periodontal disease; their high level in the GCF reflects an extent of destruction in periodontal tissues, though IL-21 has a particular significance, as it could be used to differentiate between periodontitis and gingivitis [250]. Additionally, TNF-α present in the GCF could be used for the diagnosis of periodontal diseases/inflammation, as its increased concentration in GCF was remarked in patients suffering from periodontal diseases at different stages [251].

According to a recent systematic review and meta-analysis, it was concluded that the salivary biomarkers that have a potential for the diagnosis of periodontal diseases are TNF-α, TNF-β, IL-1α, IL-1β, IL-4, IL-6, IL-8, IL-10, IL-17, IL-32, PGE2, MMP-8, MMP-9, MIP-1α, and TIMP-2. In addition, the IL-1β, TNF-α, MMP-8, and MMP-9 biomarkers could be used to monitor the prognosis of the periodontal condition after the scaling and the root planning [252].

Such findings reveal the potentiality of using the patient’s saliva for the diagnosis/prognosis of periodontal diseases relying on salivary/GFC biomarkers. This requires the assistance of digital microfluidics for the clinical translation of such promising biomarkers, allowing some chair-side Lab-on-a-chip technology available for easy and rapid clinical use [248,253,254].

### 7.2. Role of Immunomodulation as a Therapeutic Strategy in Periodontitis

Periodontitis has multiple etiological factors acting at multiple aspects, primarily the presence of dysbiotic microbial communities and the environmental and systemic health status that direct the host response to such a challenge. Periodontitis is widely accepted now to be a dysbiotic inflammatory disease; thus, the main factor affecting the extent of the destruction is now believed to be the host immuno-inflammatory status, whether hypo- or hyperresponsive to the existing dysbiotic microbiota [255]. Treatment methods employed currently fail to address the uncontrolled host immune response; hence, considerable attention is now directed toward the potential role of modulating the innate immune response to periodontal pathogens to control the inflammatory response, control osteoclastogenesis, and restore physiological bone turnover and homeostasis [256]. Immunotherapies aim to target the key players in periodontitis, particularly neutrophils, monocytes, macrophages, T lymphocytes, and inflammatory cytokines. Several strategies are now widely investigated, including the use of antioxidants to reduce oxidative stress and prevent periodontitis. Resveratrol, an antioxidant supplement, was shown to reduce the production of ROS by human gingival fibroblasts and improved the cellular response in vitro [257]. Other strategies include drugs targeting key immune cells and cytokines in periodontitis, antibacterial therapies through vaccinations, employing stem cell therapy and cell-free therapies using secretome and exosomes, gene therapies, and others (reviewed in [256]), or the use of biomaterials functionalized/loaded with immunomodulating agents (reviewed in [258]). However, the translation of these approaches clinically is rather still limited, and more research is needed to fully assess the efficacy, safety, and employment of different immunotherapies.

## 8. Conclusions

Periodontitis is an inflammatory disease that is initiated by dysbiotic oral microbiota which is believed to protect their existence through dysregulation of the host immune response. When the immune cells become unable to control the dysbiotic microbial attack, the extensive release of cytokines, matrix-degrading enzymes, and chemokines occur. Subsequently, this leads to increased periodontal tissue breakdown. Several proinflammatory cytokines released during the process of periodontitis could have a great toll on the alveolar bone. Selective inhibition of these cytokines or their cellular receptors can be beneficial to reduce periodontitis-associated bone resorption. Microbe-associated molecular patterns have a great role in initiating host immune response and bone resorption; therefore, addressing the exact mechanisms of their effect and how to counteract them is mandatory. Consequently, tissue-specific host immunity should be the future of research for the pathogenesis of periodontitis to unveil such a complicated process and hence to reach tissue-specific diagnostic/therapeutic solutions for patients with periodontal diseases worldwide.

## Figures and Tables

**Figure 1 ijms-24-04599-f001:**
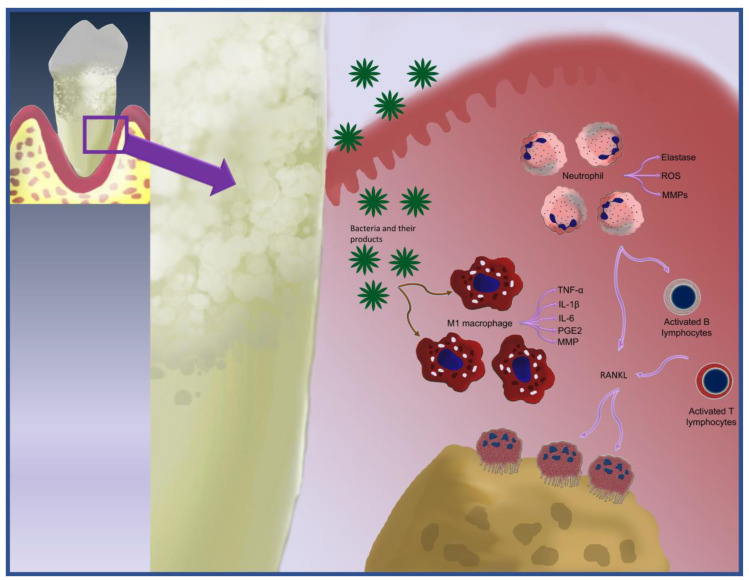
The role of inflammatory cells in periodontal tissue destruction.

**Figure 2 ijms-24-04599-f002:**
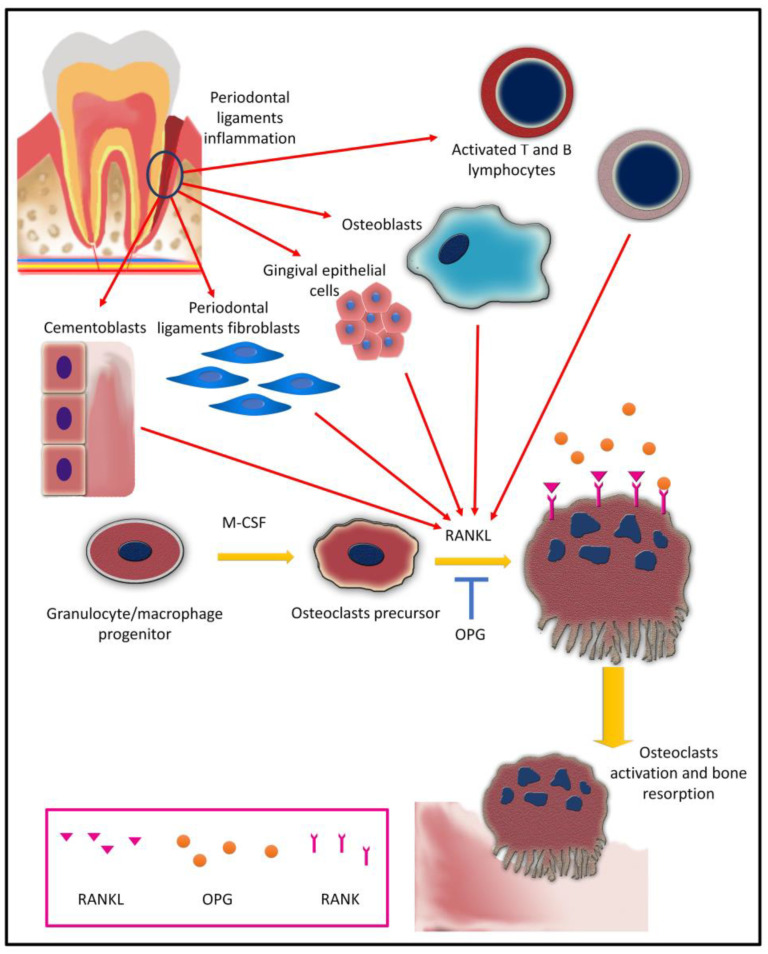
Schematic diagram representing osteoclast differentiation in periodontal diseases.

**Table 1 ijms-24-04599-t001:** Characteristics of bone cells.

Cell	Characteristics	Express	Biological Activity
**Osteoblast**	Cuboidal in shape, they are found on a bone surface, they contact each other by adherens and gap junctions	-Alkaline phosphatase -Osteocalcin -Osteopontin -Bone sialoprotien -Collα1 -Osterix -RUNX2	-They synthesize the extracellular bone matrix and promote its mineralization by matrix vesicles.
**Osteocyte**	Stellate shape, the cell body lies in lacunae and cell processes run in the canaliculi in all directions connected with gap junction	-Insulin-like growth factor-I -*c-fos* -RUNX2 -Bone sialoprotien	-They are involved in bone turnover. -Widely spread interconnected cells allow the diffusion of substances through the bone. -They act as mechanoreceptors of bone.
**Osteoclast**	Large multinucleated phagocytic cells	-Tartrate-resistant acid phosphatase -Carbonic anhydrase II -Vacuolar proton ATPase -Vitronectin receptor -Calcitonin receptor	-They are involved in bone resorption (bone remodeling) during growth or changing mechanical stresses. -They participate in the long-term maintenance of blood calcium homeostasis.

**Table 2 ijms-24-04599-t002:** Cytokines and pathways involved in osteoblastogenesis and osteoclastogenesis in periodontitis.

Cytokines	Osteoblastogenesis	Osteoclastogenesis
	Low doses enhance osteoblastogenesis via	High doses decrease osteoblastogenesis via	Enhance osteoclastogenesis via	Inhibit osteoclastogenesis via
**IL-1**	-Noncanonical Wnt-5a/Ror2 pathway.-BMP/Smad signaling pathway.	-Activating NF-κB and MAPK signaling pathway.-Suppressing BMP/signaling.-Upregulating the Wnt signaling pathway antagonists DKK1 and sclerostin.	-Potentiating RANKL via p38 MAPK pathway.-Downregulation of OPG mRNA expression via ERK pathway.-Upregulating transcriptional factors, including JNK, P38, and ERK.-Inducing osteoclast-specific genes, such as OSCAR and TRAP-Upregulating IL-1 and IL-1RI expression via c-Fos and NFATc1.	
**IL-6**	-Upregulation of RUNX2 and ALP activity through insulin-like growth factor.-Activation of transcription (STAT3) pathway.	-Downregulation of RUNX2, osterix (OSX), and OC via activation of Src homology 2 (SHP2)/MAPK/extracellular signal-regulated kinase (ERK), Janus kinase (JAK)/STAT3, and SHP2/phosphoinositide 3-kinase (PI3K)/Akt2 signaling pathways.	-Upregulating RANKL expression on osteoblasts-GP130 signaling pathway.-With high level of RANKL upregulated osteoclast differentiation through upregulation of NF-κB, ERK, and JNK phosphorylation.	-Suppression of NF-κB pathways.-With low level of RANKL, IL-6, and sIL-6R downregulated osteoclast differentiation.
**IL-17**	-Upregulating ALP, OC, RUNX2, and RANKL expression.	-Inhibiting BMP-2-induced Osteoblastogenesis with reduced expression of ALP, OC, RUNX2, and OSX expression.-Activation of ERK1,2 and c-Jun N-terminal kinase (JNK) MAPK.	-RANKL upregulation through triggering receptor expressed on myeloid cells-1 (TREM-1).	
**TNF-α**	-NF-κB through increasing expression of BMP-2, OSX, RUNX2, OC, and Wnt signaling pathway.	-Increasing production of DKK-1 and Wnt signaling pathway antagonist.	-Upregulating RANK expression on osteoclasts precursor and sensitizing precursor cells to RANKL via activation of NF-kB and SAPK/JNK.-Inhibiting WNT signaling pathway.	
**IL-1β and TNF-α**	-No change.	-Boosting the canonical Wnt/-catenin pathway and blocking the noncanonical Wnt/Ca2+ pathway.	
**IFN-γ**	-Enhancing stem cells’ antigen-presenting activities, minimizing their lysis.-Inducing bone marrow MSCs secrete functional indoleamine 2,3-dioxygenase and IL-10.	-T cell activation.	
**TGF-β**	-Canonical Smad-dependent and non-canonical Smad-independent pathways (e.g., p38, MAPK).-Forming a complex with Smad4, which interacts with RUNX2, resulting in numerous osteogenic genes to be activated-Activating Smad3 and enhancing BMP-2 production in bone marrow MSCs.	-Modifying Smad3 binding sites cause BMP-2 suppression on its promoter-Elevating tomoregulin-1levels in mice, causing BMP-2 suppression.	
**IL-10**	-Activating oxidative phosphorylation.		
**IL-23**	-Canonical Wnt-catenin pathway.		
**IL-27**	-Inhibiting osteoblast death.		
**IL-33**		-Potentiating RANKL via ERK and p38 MAPK.-Activating spleen associated tyrosine kinase, phospholipase Cc2, Grb2-associated-binding protein 2, MAPK, TAK-1, NF-kB. IL-33, TNF-α receptor-associated factor 6 (TRAF6), nuclear factor of activated T cells cytoplasmic 1, c-Fos, c-Src, cathepsin K, and calcitonin receptor.	-Impeding osteoclast differentiation factors, such as NFATc1.-Upregulation of osteoclasts apoptosis via upregulation of proapoptotic molecules, such as BAX, Fas, and FasL.-Lower levels are corelated with OPG expression in chronic periodontitis.
**IL-36 γ**		-Correlated with RANKL to OPG ratio.	
**IL-22**		-Upregulation of RANKL expression via the MAPK signaling pathway.	
**IL-8**		-Stimulating inflammatory cells to secrete IFN-γ, IL-17, TNF-α, IL-1β, and RANKL.	
**IL-11**		-Upregulating RANKL expression by osteoblasts.

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
