# Peer review of "Molecular Basis beyond Interrelated Bone Resorption/Regeneration in Periodontal Diseases: A Concise Review"

_ijms, 2023, doi:10.3390/ijms24054599_

Round 1

Reviewer 1 Report

This Concise Review is well written, systematically represents the molecular basis of the influence of periodontal disease to the bone tissue and represent very good material for all dealing the periodontal disease but also those dealing with the basic research study, especially regarding the study towards finding potential molecular targets for the therapy. Although the review is generally well written, some corrections are needed such as:

1) The names of the microorganisms have to be in italics (e.g. P. gingivalis or E. coli should be P. gingivalis and E. coli).

2) In Section name "4.2. IL-1β/TNF-α" it should be separated these two cytokines because in this way somebody may think that these two are the same and they are not. It should be changed to "IL-1β and TNF-α" or this section should come after the section 4.6. Also, at other places in the text IL-1β/TNF-α should be separated (IL-1β and TNF-α).

3) Section "4.7. TNF-α and LPS" should be connected to the section 4.6. since it describes the regulation of TNF-α by LPS and this is logical continuation to the section 4.6.

4) The "osteoblastgenesis" should be corrected to "osteoblastogenesis" (letter "o" is missing in all places where this word is mentioned in the manuscript).

5) Figure 1 should be referenced in the text and not in the section name. Also, Figure 1 does not have the legend. This should be corrected and added.

6) "P. gingivalis-LPS and Escherichia coli-LPS" should be rewritten as the following:  "P. gingivalis-derived LPS and E. coli-derived LPS".

There are numerous spelling and technical errors that should be carefully checked and corrected, but, overall I do think that this review is valuable source of information.

Author Response

Reviewer 1

Comment #1

“The names of the microorganisms have to be in italics (e.g. P. gingivalis or E. coli should be P. gingivalis and E. coli)”

Answer

Based on reviewer’s comment, the names of the microorganisms have been checked through the manuscript and corrected.

Comment #2

“In Section name "4.2. IL-1β/TNF-α" it should be separated these two cytokines because in this way somebody may think that these two are the same and they are not. It should be changed to "IL-1β and TNF-α" or this section should come after the section 4.6. Also, at other places in the text IL-1β/TNF-α should be separated (IL-1β and TNF-α).”

Answer

Following the reviewer’s valuable suggestion, the section name "4.2. IL-1β/TNF-α" has been changed into "IL-1β and TNF-α" and in other places in the text.
The section name "4.2. IL-1β/TNF-α" was transferred after the section 4.5. TNF-α

Comment #3

“Section "4.7. TNF-α and LPS" should be connected to the section 4.6. since it describes the regulation of TNF-α by LPS and this is logical continuation to the section 4.6.”

Answer

The reviewer’s comment is sincerely appreciated. Accordingly, Section "4.7. TNF-α and LPS" was connected to the section 4.5. TNF-α.

Comment #4

“The "osteoblastgenesis" should be corrected to "osteoblastogenesis" (letter "o" is missing in all places where this word is mentioned in the manuscript).”

Answer

Following the reviewer comment, the "osteoblastgenesis" was corrected to "osteoblastogenesis" in all places in the manuscript.

Comment #5

“Figure 1 should be referenced in the text and not in the section name. Also, Figure 1 does not have the legend. This should be corrected and added.”

Answer

Thanks for pointing this out, Figure 1 was referenced in the text, also, Figure 1 legend has been added.

Comment #6

“"P. gingivalis-LPS and Escherichia coli-LPS" should be rewritten as the following:  "P. gingivalis-derived LPS and E. coli-derived LPS".”

Answer

Based on reviewer’s comment, the names of the microorganisms have been checked through the manuscript and corrected.

Comment #7

There are numerous spelling and technical errors that should be carefully checked and corrected,

Answer

Thanks for pointing this out, the spelling and technical errors have been checked and corrected.

Reviewer 2 Report

I think, that this review is a narrative review and this should be mentioned in the headline or at least in the abstract. I would recommend to shorten the introduction section. Instead of describing al signal pathways I would recommend to include an additional figure that shows the effects of (for example) Elastase, RANKL and B-Cells.Instead of paragraph 2 Bone Cells, why don't the authors include a clearly arranged table including the three bone-cells and their characteristics?

Overall, the text passages should be shortened and some signaling pathways or modes of action of cytokines should be supplemented by tables or figures.

This review is nicely written, but needs some corrections. And I am still confused about the intention of the authors. This is still an overview of the molecular mechanisms in periodontal disease and incluence on the bone. However there should be a paragraph with potential clinical targets or medicamental targets in the manuscript. The authors state that host immunity should be the future of research. Do any studies about this exist? There should be a small paragraph at the end before conclusion section.

Author Response

Reviewer #2

Comment #1

“I think, that this review is a narrative review and this should be mentioned in the headline or at least in the abstract.”

Answer

Based on the reviewer’s valuable suggestion, the type of the review “narrative review” was mentioned in the abstract.

Revised text

This narrative review elaborates the important interactions between inflammatory stimuli during periodontal diseases, bone cells, MSCs, and subsequent bone regeneration or bone resorption.

Comment #2

“I would recommend to shorten the introduction section. Instead of describing all signal pathways I would recommend to include an additional figure that shows the effects of (for example) Elastase, RANKL and B-Cells. Instead of paragraph 2”

Answer

The authors agree with the reviewer; the introduction section has been summarized. A figure has been performed for describing the signal pathways Elastase, RANKL and B-Cells.

Revised text

  1. Introduction

Periodontium is considered the mirror that reflects many of the human body's internal status and secrets [1]. Periodontal diseases are chronic infectious conditions causing irreversible damage in the periodontium and its surrounding bone. Worldwide, severe periodontitis is  considered among   the most prevalent chronic inflammatory conditions affecting the periodontal apparatus. [2]. Many factors are involved inthe pathogenesis of the periodontal diseases where, the primary etiological factor is the microbial biofilm. Other etiological factors include some environmental factors, such as smoking, certain systemic diseases like diabetes mellitus, in addition to genetic risk factors affecting the inflammatory and immunologic response of the host [3]. The initiation and progression of periodontitis relies mainly on the relation/balance between the local microbiota and the immune response of the host [4].

Periodontitis is principally initiated and sustained via the dental plaque, oral microbial biofilm, in addition to the dysbiosis occurring in the periodontium and causing an extensive release of cytokines, chemokines and some matrix-degrading enzymes originating from the gingival tissues, the infiltrating immune cells, and the fibroblasts. The net effects of such destructive mediators lead to tissue breakdown, loss of periodontal attachment, and subsequently complete tooth loss [5].

The acute innate immune response during periodontal inflammation depends on a variety of immune cells, such as natural killer cells, neutrophils and macrophages, besides other variable cytokines. Neutrophils represent the first wave of immune cells within 24 h after the microbial attack. Monocyte-macrophages lineage cells represent the second wave of inflammatory cells peaking at 48–96 h [6].

In an attempt to damage the proteins, present in the membranes of some bacteria, neutrophils release elastase, however, this results in the breakdown of type I–IV collagen and the elastin of the periodontal ligaments. Elastase is also able to degrade extracellular matrix (ECM) by accelerating matrix metalloproteinases (MMPs) cascades, all of which result in attachment loss and pocket formation [7]. The breakdown of the tissues is further exacerbated by neutrophils through the release of  MMPs and reactive oxygen species (ROS)[8]. In addition, activated neutrophils increase  the expression of receptor activator of nuclear factor kappa-B ligand (RANKL),   resulting in a stimulation of the osteoclastogenesis and subsequently bone resorption [8,9]. Activated neutrophils release chemokines a mediating a chemotactic recruitment of T helper (TH)-1 and TH17 cells. TH17 augment the recruitment, the activation, and hence, the survival of neutrophils through the release of cell-derived cytokines, for instance; interleukin (IL)-17, interferon-γ (IFNγ), CXC-chemokine ligand 8 (CXCL8; also known as IL-8), granulocyte–macrophage colony-stimulating factor (GM-CSF), and tumor necrosis factor (TNF). Activated neutrophils also secret cytokines capable to mediate the survival, differentiation, and maturation of B cells [8].

Activated B and T cells are also a source of RANKL resulting in the resorption of bone present in the area of diseased gingival tissues.  Additionally, when the B cells are activated, it results in proliferation, differentiation, and maturation of plasma cells. Plasma cells can produce some cytokines such as TNF-α, transforming growth factor-β (TGF-β), IL-6, and IL-10. TNF-α induces the expression of MMPs, thus promoting MMPs-mediated periodontal tissue destruction [10].

Once, the  monocyte-macrophage system is activated by the periodontal bacteria and their products  a production of  a huge  pro-inflammatory cytokines occurs [11].  Two distinct phenotypes of macrophages exist; the pro-inflammatory (M1) macrophages and the anti-inflammatory (M2),  arising from the polarization [6]. The M1 and M2 transition is a significant mechanism for the transition between the active periodontitis and the inactive type. Macrophage polarization into M1 is triggered by various stimuli, including microbial stimuli such as IFN-γ and lipopolysaccharide (LPS). M1 produce various pro-inflammatory factors and represents the primary sources of IL-6, IL-1β, TNF-α, prostaglandin E2 (PGE2), MMP-1 and MMP-2, resulting in the degradation and destruction of the periodontal connective tissues followed by bone resorption [11].  Figure 1 summarizes the role of inflammatory cells in periodontal tissue destruction. 

Thus, the relationship between the immune surveillance; which is the processes permitting the cells of the immune system to look for and identify foreign pathogens; and the oral microbe-induced host immune response dictates the progress of the periodontal disease. If a mild host immune response and a local stimulation are balanced, there will be an immunological surveillance with a proper immune response. Though, if the pathogenicity of the local microbiota increases   due to the colonization of keystone pathogens that over-activate the host immune response, an initiation of tissue destruction arises. Yet, the detailed pathogenesis and the immune response to periodontitis remains uncertain  and still have many controversies [12].

Figure 1: the role of inflammatory cells in periodontal tissue destruction

Comment #3

Bone Cells, why don't the authors include a clearly arranged table including the three bone-cells and their characteristics?

Answer

Based on the reviewer’s valuable suggestion, a table has been added including the three bone-cells and their characteristics?

Revised text

Table 1. Characteristics of bone cells

Cell

Characteristics

Express

Biological activity

Osteoblast

Cuboidal in shape, they are found on a bone surface,  they contact each other by adherens and gap junctions

-Alkaline phosphatase

-osteocalcin

-Osteopontin

-Bone sialoprotien

-Collα1

-Osterix

- RUNX2

They synthesize the extracellular bone matrix and promote its mineralization by matrix vesicles.

Osteocyte

Stellate shape, The cell body lies in lacunae and cell processes run in the canaliculi in all directions connected with gap junction.

-Insulin-like growth factor-I

-c-fos 

-RUNX2

-Bone sialoprotien

-They are involved in bone turn over.

-Widely spread interconnected cells allow the diffusion of substances through the bone.

-They act as mechanoreceptors of bone

Osteoclast

Large multinucleated phagocytic cells

-Tartrate-resistant acid phosphatase

-carbonic anhydrase II

- vacuolar proton ATPase

-vitronectin receptor

-calcitonin receptor

-They are involved in bone resorption (bone remodeling) during growth or changing mechanical stresses

-They participate in the long-term maintenance of blood calcium homeostasis.

Comment #4

Overall, the text passages should be shortened and some signaling pathways or modes of action of cytokines should be supplemented by tables or figures.

Answer

The authors agree with the reviewer; the introduction section has been summarized and the involved cytokines in osteoblastogenesis and osteoclastogenesis have been tabulated in Table 2.

Revised text

Table 2: Cytokines and pathways involved in osteoblastogenesis and osteoclastogenesis in periodontitis

Cytokines

Osteoblastogenesis

Osteoclastogenesis

Low doses enhance Osteoblastogenesis via

High doses decrease Osteoblastogenesis via

Enhance osteoclastogenesis via

Inhibit osteoclastogenesis via

IL-1

-Non-canonical Wnt-5a/Ror2 pathway.

-BMP/Smad signaling pathway.

-Activating NF-κB and MAPK signaling pathway.
-Suppressing BMP/signaling.
-Upregulating the Wnt signaling pathway antagonists DKK1 and sclerostin.

-Potentiating RANKL via p38 MAPK pathway.

-Down regulation of OPG mRNA expression via ERK pathway.

-Upregulating transcriptional factors including JNK, P38, and ERK.

- Inducing osteoclast- specific genes such as OSCAR and TRAP

-Upregulating IL-1 and IL-1RI expression via c-Fos and NFATc1.

IL-6

-Upregulation of RUNX2 and ALP activity through insulin-like growth factor.

-Activation of transcription (STAT3) pathway.

-Downregulation of RUNX2, osterix (OSX), and OC via activation of Src homology 2 (SHP2)/ MAPK /extracellular signal-regulated kinase (ERK), Janus kinase (JAK)/STAT3, and SHP2/ phosphoinositide 3-kinase (PI3K)/Akt2 signaling path-ways.

- Upregulating RANKL expression on osteoblasts

- GP130 signaling pathway.

- With high level of RANKL upregulated osteoclast differentiation through upregulation of NF-κB, ERK, and JNK phosphorylation.

- Suppression of NF-κB pathways.

- With low level of RANKL, IL-6, and sIL-6R downregulated osteoclast differentiation.

IL-17

-Upregulating ALP, OC, RUNX2, and RANKL expression.

-Inhibiting BMP-2-induced Osteoblastogenesis with reduced expression of ALP, OC, RUNX2 and OSX expression.

- Activation of ERK1,2 and c-Jun N-terminal kinase (JNK) MAPK.

- RANKL upregulation through triggering receptor expressed on myeloid cells-1 (TREM-1).

TNF-α

-NF-κB through increasing expression of BMP-2, OSX, RUNX2, OC, and Wnt signaling pathway.

-Increasing production of DKK-1and Wnt signaling pathway antagonist.

-Upregulating RANK expression on osteoclasts precursor and sensitize precursor cells to RANKL via activation of NF-kB and SAPK/JNK.

- Inhibiting WNT signaling pathway.

IL-1β and TNF-α

-No change.

-Boosting the canonical Wnt/-catenin pathway and blocking the noncanonical Wnt/Ca2+ path-way.

IFN-γ

-Enhancing stem cells' antigen-presenting activities, minimizing their lysis.

- Inducing bone marrow MSCs secrete functional indoleamine 2,3-dioxygenase and IL-10.

- T cell activation.

TGF-β

-Canonical Smad-dependent and non-canonical Smad-independent pathways (e.g., p38, MAPK).

- Forming a complex with Smad4, which interact with RUNX2, resulting in numerous osteogenic genes to be activated

-Activating Smad3, and enhancing BMP-2 production in bone marrow MSCs.

-Modifying Smad3 binding sites cause BMP-2 suppression on its promoter. -Elevating tomoregulin-1levels in mice, causing BMP-2 suppression.

IL-10

-Activating oxidative phosphorylation.

IL-23

-Canonical Wnt-catenin pathway.

IL-27

-Inhibiting osteoblast death.

IL-33

- Potentiating RANKL via ERK and p38 MAPK.

- Activating spleen associated tyrosine kinase, phospholipase Cc2, Grb2- associated-binding protein 2, MAPK, TAK-1, NF-kB. IL-33, TNF-α receptor-associated factor 6 (TRAF6), nuclear factor of activated T cells cytoplasmic 1, c-Fos, c-Src, cathepsin K, and calcitonin receptor.

-Impeding osteoclast differentiation factors such as NFATc1.

-Upregulation of osteoclasts apoptosis via upregulation of pro-apoptotic molecules as such BAX, Fas, and FasL.

- Lower levels are  corelated with OPG expression in chronic periodontitis.

IL-36 γ

-Correlated with RANKL to OPG ratio.

IL-22

-Upregulation of RANKL expression. via the MAPK signaling pathway.

IL-8

-Stimulating inflammatory cells to secrete IFN-γ, IL-17, TNF-α, IL-1β, and RANKL.

IL-11

-Upregulating RANKL expression by osteoblasts.

Comment #5

This review is nicely written, but needs some corrections. And I am still confused about the intention of the authors. This is still an overview of the molecular mechanisms in periodontal disease and incluence on the bone. However there should be a paragraph with potential clinical targets or medicamental targets in the manuscript.

Answer

The authors sincerely appreciate the reviewer’s comment and a new section has been added
“7. Clinical implications”

Revised text

  1. Clinical implications

7.1. The potentiality of using the GCF and the salivary biomarkers as diagnostic tools for periodontal diseases

Some biomarkers could be used as diagnostic tools for the course of many diseases in the human body, including periodontal diseases [246]. Many studies were performed to correlate between the presence of; one or group of; cytokines in the saliva or the GCF, and the periodontal condition of the patient [139,244,247-249]. Focusing on the biomarkers present in the GCF, it was proved that IL-1 with its both types; IL-1α and IL-1β possess a great potential in distinguishing periodontitis from periodontal healthy cases [247]. Such biomarkers could reveal improved diagnostic ability when combined with the anti-inflammatory cytokine, IFN-γ and IL-10 [247]. Another group of biomarkers that could be used in the diagnosis of periodontal diseases are IL-1β, IL-8, MMP-13, osteoprotein, and osteoactivin [139]. As well, the level of IL-17, IL-18 and IL-21 in the GCF could be correlated with the severity of the periodontal disease, their high level in the GCF reflects an extent of destruction in the periodontal tissues. Though, the IL-21 has a particular significance, as it could be used to differentiate between periodontitis and gingivitis [250]. Additionally, TNF- α present in the GCF could be used for the diagnosis of periodontal diseases/inflammation, as its increased concentration in the GCF was remarked in patients suffering from periodontal diseases at different stages [251].

According to a recent systematic review and meta-analysis, it was concluded that the salivary biomarkers that have a potential for the diagnosis of periodontal diseases are TNF-α, TNF-β, IL-1α, IL-1β, IL-4, IL-6, IL-8, IL-10, IL-17, and IL-32, PGE2, MMP-8, MMP-9, MIP-1α, and TIMP-2. Besides, the IL-1β, TNF-α, MMP-8, and MMP-9 biomarkers could be used to monitor the prognosis of the periodontal condition after the scaling and the root planning [252].

Such findings reveal the potentiality of using patient’s saliva for the diagnosis/prognosis of periodontal diseases relying on the salivary/GFC biomarkers. This requires the assistance of the digital microfluidics for the clinical translation of such promising biomarkers, allowing some chair side Lab-on-a-chip technology available for an easy and rapid clinical use [248,253,254].

7.2. Role of immunomodulation as a therapeutic strategy in periodontitis

Periodontitis has multiple etiological factors acting at multiple aspects, primarily the presence of dysbiotic microbial communities and the environmental and systemic health status that direct the host response to such a challenge. As periodontitis is widely accepted now to be a dysbiotic inflammatory disease, thus, the main factor affecting the extent of destruction is now believed to be the host immuno-inflammatory status, whether hypo- or hyperresponsive to the existing dysbiotic microbiota [255]. Treatment methods employed currently failed to address the uncontrolled host immune response, hence, considerable attention is now directed towards the potential role of modulating the innate immune response to periodontal pathogens to control the inflammatory response, control osteoclastogenesis, and restore physiological bone turnover and homeostasis [256]. Immunotherapies aim to target the key players in periodontitis particularly neutrophils, monocytes, macrophages, T lymphocytes, and inflammatory cytokines. Several strategies are now widely investigated including the use of antioxidants to reduce oxidative stress and prevent periodontitis. Resveratrol, an antioxidants supplement, was shown to reduce the production of ROS by human gingival fibroblasts and improved the cellular response in vitro [257]. Other strategies include drugs targeting key immune cells and cytokines in periodontitis, antibacterial therapies through vaccinations, employing stem cell therapy and cell-free therapies using secretome and exosomes, gene therapies and others (reviewed in [256]), or the use of biomaterials functionalized/loaded with immunomodulating agents (reviewed in [258]). However, the translation of these approaches clinically is rather still limited, and more research is needed to fully assess the efficacy, safety, and employment of different immunotherapies.

Comment #6

The authors state that host immunity should be the future of research. Do any studies about this exist? There should be a small paragraph at the end before conclusion section.

Answer

Based on the reviewer’s comment a new section has been added before the conclusion entitled “7.2. Role of immunomodulation as a therapeutic strategy in periodontitis”

Revised text

7.2. Role of immunomodulation as a therapeutic strategy in periodontitis

Periodontitis has multiple etiological factors acting at multiple aspects, primarily the presence of dysbiotic microbial communities and the environmental and systemic health status that direct the host response to such a challenge. As periodontitis is widely accepted now to be a dysbiotic inflammatory disease, thus, the main factor affecting the extent of destruction is now believed to be the host immuno-inflammatory status, whether hypo- or hyperresponsive to the existing dysbiotic microbiota [255]. Treatment methods employed currently failed to address the uncontrolled host immune response, hence, considerable attention is now directed towards the potential role of modulating the innate immune response to periodontal pathogens to control the inflammatory response, control osteoclastogenesis, and restore physiological bone turnover and homeostasis [256]. Immunotherapies aim to target the key players in periodontitis particularly neutrophils, monocytes, macrophages, T lymphocytes, and inflammatory cytokines. Several strategies are now widely investigated including the use of antioxidants to reduce oxidative stress and prevent periodontitis. Resveratrol, an antioxidants supplement, was shown to reduce the production of ROS by human gingival fibroblasts and improved the cellular response in vitro [257]. Other strategies include drugs targeting key immune cells and cytokines in periodontitis, antibacterial therapies through vaccinations, employing stem cell therapy and cell-free therapies using secretome and exosomes, gene therapies and others (reviewed in [256]), or the use of biomaterials functionalized/loaded with immunomodulating agents (reviewed in [258]). However, the translation of these approaches clinically is rather still limited, and more research is needed to fully assess the efficacy, safety, and employment of different immunotherapies.